# Farnesol Emulsion as an Effective Broad-Spectrum Agent against ESKAPE Biofilms

**DOI:** 10.3390/antibiotics13080778

**Published:** 2024-08-17

**Authors:** Li Tan, Rong Ma, Adam J. Katz, Nicole Levi

**Affiliations:** Department of Plastic and Reconstructive Surgery, Wake Forest University School of Medicine, Winston-Salem, NC 27157, USA; ltan@wakehealth.edu (L.T.); akatz@wakehealth.edu (A.J.K.)

**Keywords:** farnesol, *Enterococcus faecium*, *Klebsiella pneumoniae*, *Enterobacter cloacae*, biofilms, burn wounds, antimicrobial resistance

## Abstract

The family of ESKAPE pathogens is comprised of *Enterococcus faecium, Staphylococcus aureus, Klebsiella pneumoniae, Acinetobacter baumannii, Pseudomonas aeruginosa*, and *Enterobacter*. Together they are the main contributors of nosocomial infections and are well established for their ability to “escape” antibiotics. Farnesol is an FDA-approved cosmetic and flavoring agent with significant anti-biofilm properties. In a proprietary emulsion, farnesol has been shown to be capable of disrupting *S. aureus, P. aeruginosa*, and *A. baumannii* biofilms. The current work demonstrates that this farnesol emulsion reduces the number of viable bacteria, while also leading to reductions in biomass, of the other three ESKAPE pathogens: *Enterococcus faecium, Klebsiella pneumoniae,* and *Enterobacter,* both in vitro and in an ex vivo human skin model. A concentration of 0.5 mg/mL was effective for impeding biofilm development of all three bacteria, while 1 mg/mL for *E. faecium* and *K. pneumoniae*, or 0.2 mg/mL for *E. cloacae,* was able to kill bacteria in established biofilms. Contrary to antibiotics, no resistance to farnesol was observed for *E. faecium* or *K. pneumoniae*. The results indicate that farnesol is effective for direct cell killing and also has the ability to induce biofilm detachment from surfaces, as confirmed using Live/Dead image analysis. Our findings confirm that farnesol emulsion is an effective broad-spectrum agent to impede ESKAPE biofilms.

## 1. Introduction

The widespread use of antibiotics has provoked the emergence of multidrug-resistant (MDR) bacteria, which are associated with a progressive increase in nosocomial infections worldwide, placing a significant burden on healthcare systems and posing one of the greatest threats to human health [1]. Among them, six notorious MDR superbugs, including two Gram-positive (G+) bacteria, *Enterococcus faecium* and *Staphylococcus aureus*, plus four Gram-negative (G−) bacteria, *Klebsiella pneumoniae*, *Acinetobacter baumannii*, *Pseudomonas aeruginosa*, and *Enterobacter* species, have been acronymically dubbed as “ESKAPE” pathogens due to their capability of “escaping” susceptibility to almost all available antibiotics [2]. These bacteria are dominant causes of life-threatening infections throughout the world [3], and the World Health Organization has listed each ESKAPE pathogen as a critical (the four G− bacteria) or high (the two G+ species) priority pathogen, for which new antibiotics/treatments are urgently needed [4].

Biofilms, the structured, surface-attached microbial communities encased in a self-produced polymeric matrix [5], instigate more than 80% of human microbial infections [6]. The capacity of ESKAPE pathogens to form biofilms on abiotic surfaces such as medical devices and catheters, as well as on host tissues like respiratory, urinary, and gastrointestinal tract mucosa, significantly contributes to persistent infections [7]. Multiple biofilm-associated factors have been found to contribute to the increased antibiotic resistance of biofilms, including the restriction of antibiotic penetration through the biofilm matrix, the secretion of antibiotic-modifying enzymes, lower metabolic activity, the biofilm-mediated upregulation of bacterial efflux, and increased gene exchange and mutation frequency [8]. Currently, there is no agent commercially available for either the prevention or treatment of ESKAPE biofilm-associated infections.

With the development of the SARS-CoV-2 pandemic, new challenges to managing ESKAPE pathogens have arisen. The virus is still prevalent, and patients with viral infection are 6.9% more likely to have ESKAPE pathogen infections, especially Gram-negative, due to their already compromised immune system [9]. The use of more antibiotics, as well as alcohol-based hand sanitizers, as needed to fight the virus, has further promoted a tendency toward greater antimicrobial resistance [10,11]. Interestingly, at the onset of the SARS-CoV-2 pandemic, antimicrobial resistance rates rose, most likely due to the increased use of antibiotics to combat secondary infections; however, resistance rates decreased in 2022 as the pandemic began to resolve [12]. Unfortunately, the trend of the increasing antimicrobial resistance of ESKAPE pathogens is continuing to increase [11], necessitating new approaches to tackle the biofilms without the potential of developing resistance to anti-biofilm agents.

Farnesol is a component of essential oils from fruits, vegetables, and herbs and has an acyclic sesquiterpene alcohol structure [13]. The Food and Drug Administration views farnesol as a GRAS (generally recognized as safe) compound, allowing for application at high concentrations in foods and cosmetics [14,15,16]. Farnesol has been demonstrated to be an anti-fungal biofilm inhibitor [17], in addition to having applications as an antimicrobial, antitumoral, neuroprotective, hepaprotective, and cardioprotective agent [18,19,20,21,22]. It has further been demonstrated that farnesol can boost the efficacy of antibiotics [23,24,25,26]. We have recently developed a proprietary emulsion of farnesol and revealed that it is highly effective for the biofilm prevention and treatment of *A. baumannii*, *S. aureus*, and/or *P. aeruginosa*, three members of the ESKAPE pathogens [27,28].

The current work evaluated the anti-biofilm properties of the farnesol emulsion against the rest of the ESKAPE pathogens: *E. faecium*, *K. pneumoniae*, and *Enterobacter*. We chose *E. cloacae* as a representative of the whole *Enterobacter* species because *E. cloacae* is the most frequently isolated species among the genus *Enterobacter* [29]. It is worth mentioning that both the selected *K. pneumoniae* and *E. cloacae* strains are New Delhi Metallo-beta-lactamase 1 (NDM-1)-positive bacteria, which are considered “superbugs” due to their resistance to most antibiotics [30]. We found that the farnesol emulsion is effective at both inhibiting biofilm formation and also disrupting established biofilms of *E. faecium*, *K. pneumoniae*, and *E. cloacae*, both in vitro and ex vivo. This potent farnesol emulsion is an unexpected, but highly effective, broad-spectrum anti-biofilm agent to combat ESKAPE pathogens.

## 2. Results

### 2.1. Farnesol Impedes Biofilm Development

The effect of farnesol on impeding the development of *E. faecium* biofilms was compared to the corresponding ethanol controls at increasing farnesol doses. Farnesol is highly effective for preventing biofilm formation since as low as 0.5 mg/mL of farnesol caused a more than 20,000-fold reduction in colony-forming units (CFU), while farnesol at 2 mg/mL resulted in a nearly 300,000-fold CFU reduction (Figure 1A). This result was further confirmed by visualizing *E. faecium* cells in the presence of moderate [in order to show the transition from green (live) to red (dead) fluorescence] doses (up to 0.5 mg/mL) of farnesol (Figure 1B), followed by quantitative analysis of fluorescence intensity using Photoshop^®^ (25.6.0 Release) (Figure 1C), and biomass and biofilm thickness using Comstat2 (Version 2.1 1 July 2015) (Figure 1D). Farnesol kills *E. faecium* cells in a dose-dependent manner, causing a gradual decrease in green fluorescence and a corresponding increase in red fluorescence with increased farnesol doses (Figure 1B–D).

The potential of farnesol to inhibit the formation of *K. pneumoniae* biofilm was determined, and 0.5 mg/mL was identified as optimal, with an over 70% reduction in CFUs and no inhibition of biofilm by the ethanol vehicle alone (Figure 2A). Quantitative analysis of Live/Dead viability confirmed that farnesol inhibits *K. pneumoniae* biofilm development (Figure 2B–D). *K. pneumoniae* cells were killed by farnesol starting at 0.1 mg/mL, with the number of dead cells peaking at 0.5 mg/mL (Figure 2C) or 0.2 mg/mL (Figure 2D). The concentration variance is a result of the difference in analytical metrics by Photoshop^®^ and Comstat2, since Comstat2 accounts for fluorescence from the three-dimensional biomass and excludes detached biofilm material [32]. Farnesol impedes the attachment of dying *K. pneumonia,* resulting in a minimal red signal in the images, although analytical software could detect a red signal to allow for measurement, as compared to the stronger green signal. (Figure 2B–D). As will be shown later, low doses of farnesol appear to facilitate the detachment of live bacteria from biofilm, beginning at 0.2 mg/mL, and this is the reason why low red, “dead”, signals are observed in Figure 2, since the viable and dead bacteria have been detached from the biofilm during the washing step, which further facilitates biofilm detachment.

The effects of farnesol against the biofilm formation of *E. cloacae* were also examined. Both doses (0.2 and 0.5 mg/mL) of farnesol were effective for significantly inhibiting biofilm formation, as indicated by around 80 and 90% reductions in CFUs, respectively (Figure 3A). This result was consistent with the quantitative Live/Dead viability analysis of *E. cloacae* biofilm development (Figure 3B–D). Farnesol killed *E. cloacae* cells starting at 0.1 mg/mL, with the dead (red) cells peaking at 0.2 mg/mL of farnesol, and the majority of dead cells being unable to attach to the surface at 0.5 mg/mL of farnesol (Figure 3B–D). As will be shown later, low doses of farnesol, beginning at 0.2 mg/mL, appear to facilitate the detachment of live bacteria from biofilm, and this is the reason why low red, “dead”, signals are observed in Figure 3, since the dead bacteria have been detached from the biofilm. 

### 2.2. Farnesol Disrupts Established Biofilms

Besides the inhibition of biofilm formation, farnesol was also found to disrupt established biofilms of *E. faecium*, with farnesol at 3 or 6 mg/mL inducing an over 12,000-fold CFU reduction (Figure 4A). This result was further confirmed by the viability analysis of *E. faecium*-established biofilms (Figure 4B) and their corresponding quantitative analyses (Figure 4C,D). Farnesol kills biofilm-encased *E. faecium* at concentrations as low as 1 mg/mL. Unexpectedly, the intensity of red (for dead cells) fluorescence was highest at 1 mg/mL of farnesol (Figure 4B–D), suggesting that some of the dead (red) *E. faecium* biofilm-encased cells failed to remain attached to the surface with increasing of farnesol doses.

Farnesol is also effective against established *K. pneumoniae* biofilms, with a concentration of 1 mg/mL resulting in an approximate 75% reduction in CFUs (Figure 5A). Higher doses were also effective; however, the effect of 6 mg/mL of farnesol was overshadowed by the reduction in viability due to the ethanol vehicle control (Ctrl_6 = 20% of ethanol), proposing that farnesol is not superior to high doses of ethanol (Figure 5A). As shown in Figure 5B–D, farnesol also disrupted established biofilms of *K. pneumoniae*, with doses up to 1 mg/mL reducing live (green) biofilm-encased cells and increasing dead (red) biofilm-encased cells.

We further evaluated the effect of farnesol to combat established biofilms of *E. cloacae*, finding that 0.2 mg/mL of farnesol was the best dose for disrupting established biofilms, while its vehicle (ethanol) control did not have much effect, as indicated by a more than 70% reduction in CFUs (Figure 6A). This result was further confirmed by the Live/Dead viability analysis of established biofilms of *E. cloacae* (Figure 6B) and their corresponding quantitative analyses (Figure 6C,D). Farnesol was able to kill biofilm-encased cells of *E. cloacae* in a dose-dependent fashion (within the dose range of 0 to 0.2 mg/mL), as evidenced by the decrease in green fluorescence (Figure 6B–D) and the corresponding increase in red fluorescence (Figure 6C,D). Surprisingly, unlike *E. faecium* or *K. pneumoniae*, the red signal was barely observable in the biofilm images except for the highest farnesol dose (0.2 mg/mL) (Figure 6B). Moreover, both Photoshop^®^ and Comstat2 software detected moderate red fluorescence for the ethanol control (Figure 6C,D). This could be due to (1) the difficulty of the dead biofilm-encased cells of *E. cloacae* to remain attached to the surface, but the live biofilm-encased cells might help to trap them, and (2) low doses (0.05 or 0.1 mg/mL) of farnesol, which could detach *E. cloacae* biofilm mass without killing it. One of the challenges of using farnesol is its lack of solubility for interfacing with aqueous media. Our prepared farnesol emulsion uses ethanol as an intermediate step, and the results above indicate that the ethanol vehicle is also toxic at higher concentrations (Control_0.5 = 1.7% of ethanol; Control_1 = 3.3% of ethanol; Control_3 = 10% of ethanol; Control_10 = 33% of ethanol) (Figure 2A, Figure 3A, Figure 4A, Figure 5A and Figure 6A).

### 2.3. No Resistance to Farnesol Was Observed

Bacterial resistance to antibiotics is a clinical challenge, which is especially prominent with ESKAPE infections [7], and resistance can develop with prolonged exposure. Therefore, we explored the potential for *E. faecium* to develop resistance to farnesol as compared to the antibiotic rifampicin. *E. faecium* was passaged in the presence of sub-inhibitory [1/2 × minimal inhibitory concentration (MIC)], with no statistically significant resistance to farnesol observed, even after 20 continuous passages (Figure 7A). The MIC was determined visually, as described in the methods presented in Section 4.6; thus, the statistical differences at each passage were not available. Nonetheless, the trends of increasing resistance to rifampicin and no resistance to farnesol were observed, with a profound increase in resistance to rifampicin after only two passages, leading to a >30,000-fold increase in the MIC. 

We also assessed the capacity of *K. pneumoniae* to develop resistance to farnesol. Twenty continuous passages of *K. pneumoniae* in the presence of 1/2 × MIC of farnesol did not select isolates resistant to farnesol, whereas exposure to the antibiotic rifampicin elicited a gradual increase in MIC, ultimately resulting in a 32-fold increase in the MIC after 16 passages (Figure 8A). The MIC was determined visually, as described in Section 4.6; thus, statistical differences at each passage were not available. Nonetheless, the trend for increasing resistance to rifampicin was observed, with only a negligible trend toward farnesol resistance. Collectively, the above results demonstrate that farnesol combats *E. faecium* and *K. pneumoniae* without inducing therapeutic resistance.

### 2.4. Farnesol Directly Kills Bacteria and Also Detaches Biofilms

Farnesol has been shown to directly kill *Staphylococcus aureus* by inducing membrane disruption [33]; hence, we sought to evaluate whether farnesol also kills *E. faecium* by examining the rapidity of propidium iodide (PI) influx. Established *E. faecium* biofilms exposed to farnesol indicate a rapid increase in the PI signal, which is indicative of damage to the bacterial cell membranes, resulting in cell death. As shown in Figure 7B, 3 mg/mL had the most rapid and profound effect, and the result corresponds to the CFU data from Figure 4A, which supports the idea that farnesol, at this dose, drastically reduced cell viability. Notably, ethanol alone did not elicit a similar effect, supporting the notion that it is farnesol instigating cell membrane damage that results in cell death. Similar PI influx results also occurred with *K. pneumoniae*, with 1 mg/mL of farnesol displaying the greatest influx (Figure 8B), which corresponds to the results of Figure 5, showing that 1 mg/mL of farnesol kills these bacterial cells. These results demonstrate that farnesol is able to directly kill both the Gram-positive *E. faecium* and the Gram-negative *K. pneumoniae*, perhaps by disrupting the cell membrane, similar to what happens in *S. aureus*.

Farnesol’s mechanism of action appears to involve more than cell membrane damage and cell killing, as it has also demonstrated the ability to facilitate biomass detachment without cell killing [27,28,34]. To confirm this phenomenon and determine the concentrations needed for detachment versus cytotoxicity, we examined the supernatant of *E. faecium* biofilms. As shown in Figure 7C, the ethanol vehicle does not cause a release of biomass into the supernatant, as determined by the lack of either green or red material. Farnesol at a low dose of 0.2 mg/mL resulted in green biomass material propagating into the supernatant, as indicated by the green material observed. Increasing doses of farnesol appear to break the floating biofilm material into smaller pieces, although they are still alive, as indicated by the green signal, with the highest dose of 1 mg/mL being able to kill cells in the floating biomass, as indicated by the red signal in Figure 7C. As shown in Appendix A, the centrifugation of the detached *E. faecium* biomass resulted in a pellet that could be stained for cell viability (green = live and red = dead). Biofilm treated with ethanol had a minimal green pellet, indicative that the cells were viable. Exposure to farnesol at increasing doses resulted in the development of a more dense, red pellet, indicating more dead cells. This result supports those of Figure 7C, implying that farnesol might detach (without killing), disintegrate, and then kill biofilm-encased cells with increasing farnesol doses. Similar Live/Dead viability results occurred with *K. pneumoniae* and *E. cloacae*, indicating that the farnesol doses employed (0.5 and 1 mg/mL for *K. pneumoniae*; 0.2 mg/mL for *E. cloacae*) were bactericidal (Figure 8C and Appendix A). These results are consistent with those of Figure 5 and Figure 6, demonstrating that higher doses of farnesol reduce cell viability. Overall, the results support that farnesol is bactericidal at high doses, while sub-lethal doses can facilitate biofilm disruption. 

### 2.5. Farnesol Is Effective against Biofilm-Associated Skin Infections

Vancomycin-resistant *Enterococcus* has emerged as a causative factor in burn wound infections [35]; therefore, we examined the efficiency of farnesol against *E. faecium* biofilms on ex vivo intact, or burned, human skin. *E. faecium* biofilm was inoculated onto the skin with 1 mg/mL of farnesol for 24 h to evaluate the inhibition of biofilm formation. A decrease in *E. faecium* biofilm on top of the epidermis was observed (as shown by arrowheads indicating the loose material atop the epidermis) with farnesol treatment (Figure 9A). This result was confirmed by the reduction in *E. faecium* CFUs on intact (Figure 9B) and burned (Figure 9C,D) skin from two independent donors. Although farnesol was found to be effective against established biofilms of *E. faecium* in vitro at a concentration of 3 mg/mL (Figure 4A), this dose was not effective against the established biofilms on the skin, necessitating the use of a higher dose of 15 mg/mL. Hematoxylin and eosin (H&E) staining of the skin showed that farnesol (15 mg/mL) significantly diminished 24 h old, established biofilms of *E. faecium* on intact human skin, as demonstrated by the decrease in biofilm (identified as loose material with arrowheads) on the skin surface (Figure 9E) and substantial CFU reductions (Figure 9F). Farnesol at 15 mg/mL was effective against established *E. faecium* biofilms on burned skin, as observed by the reduction in biofilm on the skin surface (Figure 9G) and CFU reductions (Figure 9H).

*K. pneumoniae* is a common Gram-negative burn wound pathogen [36]; hence, we evaluated whether farnesol was also effective against *K. pneumoniae* biofilms on ex vivo intact, or burned human skin. The ethanol vehicle did not kill *K. pneumoniae* on the ex vivo human skin; thus, higher doses of farnesol could be used (6 mg/mL for biofilm formation; 15 mg/mL for established biofilms). Farnesol at 6 mg/mL inhibited *K. pneumoniae* biofilm formation on intact human skin, as observed by less *K. pneumoniae* biofilm (indicated by the loose material highlighted by arrowheads) on the epidermis (Figure 10A) and further confirmed by the significant CFU decrease on the skin (Figure 10B). Similar results ensued for the inhibition of *K. pneumoniae* biofilm formation on burned human skin (Figure 10C,D). The established *K. pneumoniae* biofilms on intact human skin seemed to cause epidermal detachment (see the control in Figure 10E). However, farnesol at 15 mg/mL inhibited this epidermal detachment and also reduced the biofilm on the epidermis (indicated by the loose material highlighted by arrowheads) (Figure 10E). A more than 90% reduction in CFUs on the skin was observed using 15 mg/mL of farnesol (Figure 10F). Farnesol at 15 mg/mL also reduced *K. pneumoniae* biofilms on burned skin (indicated by the loose material highlighted by arrowheads) (Figure 10G), with lower CFUs (Figure 10H).

*Enterobacter* species represent additional Gram-negative pathogens in burn wounds [36]. Using an ex vivo intact or burned human skin model to evaluate biofilms of *E. cloacae*, we observed that ethanol alone did not kill *E. cloacae,* allowing for usage of higher farnesol doses (1 mg/mL for biofilm formation; 15 mg/mL for established biofilms). Similar to the above *E. faecium* case, 1 mg/mL of farnesol inhibited *E. cloacae* biofilm formation on intact human skin, as visualized by a reduction in *E. cloacae* biofilm on top of the skin surface (indicated by the loose material highlighted by arrowheads) (Figure 11A). This result was confirmed by an about 80% CFU reduction on the skin (Figure 11B). Interestingly, the same farnesol treatment appeared to be more effective at inhibiting *E. cloacae* biofilm formation on burned human skin. H&E staining of the skin biopsy showed that *E. cloacae* penetrated the stratum corneum and developed a deep pocket of biofilm underneath. In contrast, farnesol at 1 mg/mL protected epidermal integrity, with significant biofilm reduction on the epidermis (Figure 11C), which was confirmed by a 90–99% reduction in CFUs on the skin (Figure 11D). With regard to established *E. cloacae* biofilms, 15 mg/mL of farnesol showed significant anti-biofilm activity against the established biofilms on the intact skin, as demonstrated by the biofilm reduction on the skin surface (indicated by the loose material highlighted by arrowheads) (Figure 11E) and substantial CFU decreases (Figure 11F). Farnesol at 15 mg/mL was also effective at combating established *E. cloacae* biofilms on burned skin, with farnesol possibly protecting the skin from bacterial penetration (arrowheads indicate more biofilm material in the control compared to the farnesol-treated skin) (Figure 11G), along with more than a 90% CFU reduction (Figure 11H).

## 3. Discussion

The number of conventional antibiotics that are effective against ESKAPE pathogens declines annually [1], making it imperative to find alternative options to treat ESKAPE infections. Our current work shows that farnesol, a natural GRAS compound, which is commercially available as a cosmetic and flavor agent, when formulated in a proprietary emulsion using ethanol, is highly effective for both inhibiting biofilm formation and also for disrupting established biofilms of every type of ESKAPE pathogen, both in vitro and ex vivo [27,28]. Plus, there was no development of resistance to the farnesol emulsion even after prolonged culture in the presence of sub-MIC doses, reducing the concern of antibiotic resistance. 

Farnesol seems to combat ESKAPE biofilms using two important features: (1) the direct killing of ESKAPE cells, possibly via cell membrane disruption, to eradicate future recurrence, and (2) the detachment of biofilms from surfaces/tissues at sub-lethal doses to stop biofilm development and facilitate biofilm removal. ESKAPE biofilm-associated infections are common in nosocomial infections, especially in burn wound infections [37]. Burn wound management usually requires frequent debridement to clean the wound area; however, debridement is often not effective at removing ESKAPE biofilms entirely and cannot inhibit development [35]. Thus, the dual functions of both bacterial killing and biofilm detachment make farnesol an ideal candidate for incorporation into debridement irrigation solutions for enhanced wound bed preparation for healing. Polymicrobial biofilms composed of multiple species of bacteria, including multiple members of the ESKAPE pathogens, have also been identified in burn wounds [35] and further complicate clinical strategies for clearance. The use of one single agent, such as farnesol, to eliminate all ESKAPE pathogens at once would be favorable. Although the scope of the current work evaluated biofilms in burn wounds, the outcome can easily be extended to other biofilm-related chronic wounds, such as non-healing surgical site infections, venous stasis ulcers, diabetic foot ulcers, pressure ulcers, and decubitus ulcers [38,39]. 

The safety of farnesol has been well documented for its potential topical application. It is an FDA-approved GRAS compound, and high doses (up to 120 mg/mL) of farnesol have been proven to be safe on human skin as a fragrance enhancer [15]. The highest dose of farnesol that we used to combat ESKAPE biofilms (15 mg/mL) was observed histologically to be safe on ex vivo human skin, and it also protects human skin cells from ethanol killing, as we have previously demonstrated [27,28]. Additionally, farnesol (and its ethanol vehicle) is cheap compared with most commercial antibiotics, facilitating farnesol emulsion as an effective, safe, and affordable broad-spectrum agent (e.g., farnesol-related sprays, irrigation solutions, and wound dressings) for the prevention and treatment of ESKAPE biofilm-related skin infections.

Our studies have demonstrated that farnesol emulsion is effective against ESKAPE biofilms; however, there are some limitations. First, we selected MDR clinical strains for each species to evaluate clinical significance, with the assumption that, if farnesol could combat an MDR strain, it should easily conquer other drug-sensitive strains of the same species. However, the anti-biofilm efficacy of farnesol against different strains could vary. Secondly, the vehicle (ethanol) used to prepare the farnesol emulsion is inherently antimicrobial, although farnesol is often superior to ethanol alone. It is possible that ethanol might play an extra role (beyond carrying a high concentration of farnesol in our stable formulation) in improving the anti-biofilm effect of farnesol. Our attempt to evaluate the effect of farnesol alone (not in an emulsion form) failed because farnesol is oily by nature, making it difficult to disperse in aqueous media for interacting with bacteria. In addition, since farnesol has been approved as a food ingredient by the FDA, its potential clinical applications in internal medicine suggest further investigation (e.g., using aerosolized farnesol to treat lung infections or a farnesol-embedded coating on medical devices).

Currently available clinical treatments for MDR ESKAPE infections mainly depend on a combination of antibiotics [40], the use of which can promote antibiotic resistance. Thus, innovative strategies for combating ESKAPE infections are needed. The work presented here demonstrates that farnesol emulsion at the low dose of 0.5 mg/mL can prevent the in vitro development of *E. faecium, K. pneumoniae*, or *E. cloacae* biofilms. The higher concentration of 1 mg/mL was needed to disrupt established biofilms of *E. faecium* and *K. pneumonia*, although the low dose of 0.2 mg/mL could kill *E. cloacae* bacteria in established biofilms. The mechanism by which farnesol is effective appears to be two-fold. First, it can directly kill bacteria by damaging the cell wall. Secondly, low doses demonstrate a potential to cause biofilm detachment from a surface, and the disruption of established biofilms is critical for reducing infectious burdens in the clinical setting. Even though higher doses (15 mg/mL) of farnesol were needed to disrupt established biofilm on ex vivo skin, the low dose of 1 mg/mL was effective for halting the development of *E. faecium* or *E. cloacae* biofilms. In addition, the use of farnesol impeded epidermal damage. These are key results, which indicate that farnesol could support wound care management. Given the well-established safety profile of farnesol, coupled with our results that its emulsion form is effective against ESKAPE biofilms prevalent in burn wounds without inducing therapeutic resistance, this work indicates beneficial opportunities to repurpose farnesol as a novel, effective, and broad-spectrum infection control agent primed for translation into clinical applications.

## 4. Materials and Methods

### 4.1. Bacterial Strains and Culture

The bacterial strains used in this study were *E. faecium* (BAA-2316) [41], *K. pneumoniae (*BAA-2146) [42], and *E. cloacae* (BAA-2468) [43] from the American Type Culture Collection (ATCC). These strains are prevalent and clinically significant MDR strains for each species, which were selected to emphasize potential clinical significance. Brain heart infusion (BHI, Sigma, St. Louis, MO, USA) was used to culture *E. faecium*, with nutrient broth (NB, Becton Dickinson, Franklin Lakes, NJ, USA) for *K. pneumoniae* and NB containing 25 µg/mL of imipenem (NBI, Sigma) for *E. cloacae.* Frozen stock was streaked onto an agar plate and grown overnight at 37 °C. Then a single colony from the plate was inoculated into broth and cultured overnight at 37 °C, at 160 rpm. Bacteria were then centrifuged at 2500× *g* for 10 min, the pellet was suspended in fresh broth, and the optical density at 600 nm (OD600) was measured using a spectrophotometer. Bacteria were diluted to 1 × 10^6^ CFU/mL for biofilm formation or 1 × 10^8^ CFU/mL for established biofilms for exposure to farnesol or ethanol controls. CFUs/mL were determined by serial dilution, growth on agar plates at 37 °C for 18 h, and visual colony counting [44].

### 4.2. OD600 and Determining CFU/mL

The linear equation relating OD600 and CFU/mL was determined for ease of subsequent calculations for each bacterial type. First, suspended bacteria were measured to specify a starting OD600. Bacteria were then serially diluted in broth to obtain OD600 readings of ~0.1, 0.2, 0.4, and 0.8, and the number of viable bacteria for each OD600 value was determined by plating onto agar, colony development at 37 °C, and visual counting. The linear equation between OD600 and CFU/mL and coefficients of determination (R^2^) values were analyzed using GraphPad Prism 9 (version 9.2.0) for each bacterial strain.

### 4.3. Inhibiting Biofilm Formation

Apheresis-derived pooled human plasma (Innovative Research, Novi, MI, USA) was dispersed at 20% concentration in 50 mM sodium bicarbonate (Sigma), and 100 µL was added to the wells of a 96-well polystyrene plate for 24 h to develop a coating to better facilitate biofilm attachment. Biofilms were developed using overnight bacterial cultures, and then bacteria (1 × 10^6^ CFU/mL) were cultured in 100 µL of broth containing farnesol, at concentrations of 0.1, 0.2, 0.5, 1, 2, 3, or 6 mg/ mL. Farnesol was purchased from Cayman Chemical (Ann Arbor, MI, USA) dissolved in 100% ethanol at 30 mg/mL and diluted into broth immediately before exposure to the bacteria. Bacteria were exposed to broth with the same volume of 100% ethanol, but without farnesol, to serve as a control. Since ethanol was the carrier solvent for the creation of the farnesol emulsion and farnesol is not soluble in aqueous media, we did not include an additional negative control of water alone. Biofilms were developed by the incubation of the bacteria, farnesol, or ethanol for 24 h at 37 °C in a humidified container. Biofilms were then washed with 100 µL of phosphate-buffered saline (PBS) to remove planktonic cells, and biofilm-residing bacteria were dislodged by vigorous (≥10 times) pipetting with 100 µL of PBS. Serial dilutions of the biofilm-residing bacteria were prepared, plated onto agar, incubated at 37 °C, and visually counted to quantify in terms of CFU/mL. The lowest boundary of detection was 50 CFU/mL.

### 4.4. Disrupting Established Biofilms

Established biofilms were developed by culturing 250 µL of 1 × 10^8^ CFU/mL of bacteria in plasma-coated 96-well plates at 37 °C for 24 h. The planktonic cells were aspirated, and the biofilms were washed in 250 µL of PBS to remove non-biofilm-residing bacteria. Established biofilms were then exposed to 100 µL of farnesol at concentrations of 0.05, 0.1, 0.2, 0.5, 1, 2 3, 6, 10, or 15 mg/mL or the same volume of ethanol as a control, in broth at various concentrations for 24 h. Following exposure, the supernatant was aspirated, then biofilms were washed with 100 µL of PBS followed by the dislodgement of the bacteria with 100 µL of PBS using vigorous (≥10 times) pipetting. The CFU/mL of viable bacteria was quantified via serial dilution, plating onto agar, followed by incubation overnight at 37 °C to develop colonies to allow for visual counting. 

### 4.5. Viability Assay Using Live/Dead with Quantitative Microscopy 

Farnesol’s potential to inhibit biofilm formation or disrupt established biofilms was visualized and quantified using the FilmTracer™ (Invitrogen, Waltham, MA, USA) Live/Dead biofilm viability assay. Biofilm evaluation was completed in Lab-Tek™ chambered coverglass (Nunc, Waltham, MA, USA) pre-coated with human plasma. The evaluation for the inhibition of biofilm development was performed by culturing 500 µL of 1 × 10^6^ CFU/mL bacteria in broth with farnesol or the ethanol control. After 24 h of incubation, planktonic cells were removed, and biofilms were washed and prepared for staining. Alternatively, biofilms were first established using 500 µL of 1 × 10^8^ CFU/mL. Established biofilms were exposed to 500 µL of farnesol or ethanol in broth for 24 h at 37 °C; then the supernatant was removed, and biofilms were washed and stained. Developing or established biofilms were treated for 20–30 min at room temperature, in the dark, with 250 µL of water containing 10 µM of SYTO^®^ 9 green fluorescent nucleic acid stain and 60 µM of propidium iodide (PI) red fluorescent nucleic acid stain. Biofilms were then washed with water, kept hydrated, and observed using a Keyence^®^ (Keyence, Itasca, IL, USA) BZ-X800/BZ-X810 All-in-One fluorescence microscope, with z-stacks obtained to allow for the visualization of multiple biofilm planes. Fluorescence from captured images was analyzed using Photoshop^®^ to quantify the overall intensity. The same images were also analyzed using Comstat2 (www.comstat.dk) to quantify the biomass and examine the biofilm structure [32,45].

To evaluate the viability of detached cells following farnesol treatment, the supernatant containing detached cells (400 µL) was centrifuged at 17,000× *g* for 5 min; then the pellet was washed, re-suspended, and stained with 250 µL of the SYTO^®^ 9-PI mixtures, followed by an additional centrifugation to remove unbound fluorescent dyes. The stained, pelleted biomass was transferred into a plasma-coated chambered coverglass and allowed to settle at room temperature for 30 min while protected from light, followed by imaging using the Keyence^®^ microscope.

### 4.6. Evaluation of Antimicrobial Resistance

Procedures to develop antimicrobial resistance have been published previously [46], and we utilized these guidelines to evaluate if resistance to farnesol would be developed, as compared to the antibiotic rifampicin (Bedford Laboratories, Bedford, OH, USA). Five microliters of 1 × 10^8^ CFU/mL of bacteria was added to 95 µL of broth containing farnesol (0.125 to 16 µg/mL for *E. faecium*; 0.032 to 3 mg/mL for *K. pneumoniae*) or rifampicin (0.125 to 16 µg/mL) for the first passage. Bacterial suspensions were then shaken at 160 rpm and 37 °C overnight. Then, we determined the lowest concentration of farnesol or rifampicin resulting in no visible bacterial growth (MIC). Due to the extensive number of passages and the fact that the MIC could be determined visually, serial dilutions and CFUs were not quantified at each passage; hence, no statistical evaluation was possible for these datasets. The obtained 0.5-fold MIC suspension was diluted 10-fold with fresh broth, with 5 µL being added to 95 µL of farnesol or rifampicin-containing broth, and bacteria passaged during exposure to the agents. Farnesol or rifampicin concentrations were increased based on the daily MIC results, and the process was continued for 20 passages.

### 4.7. Evaluation of Cell Membrane Damage Using Propidium Iodide (PI) 

Biofilms were established in plasma-coated 96-well plates, and they were exposed to 20 µM of PI (Invitrogen, Carlsbad, CA, USA) at room temperature, in the dark, for 10 min. Then, farnesol or the ethanol control was added to reach various final concentrations in a volume of 150 µL. The plates were incubated at room temperature, in the dark, for 30 min, followed by the removal of unbound PI, washing, and the addition of 100 µL of water into each well. The fluorescence from PI was measured every 30 s for 5 min using a TECAN Infinite M200 microplate reader (Mennedorf, Switzerland). 

### 4.8. Developing and Treating Biofilms on Ex Vivo Human Skin

Abdominal skin from healthy human donors was procured from the Department of Plastic and Reconstructive Surgery at the Wake Forest University School of Medicine, under an Institutional Review Board (IRB)-approved protocol. Samples were de-identified and managed as human waste; thus, no informed consent was needed. The study authors were not involved in the procurement of the skin samples. Subcutaneous fat was removed, and three-centimeter portions of skin were prepared as sample substrates for growing biofilm. To prepare the samples, the epidermis was soaked with 70% ethanol for 5 min, then with sterile PBS for 10 min, and the washing cycle was repeated four times. Skin samples subjected to second-degree burn injury had a 3 cm diameter brass cylinder heated to 100 °C in polyethylene glycol applied for 10 s. The skin was placed (epidermal side up and only the dermis submerged) into the wells of a 6-well plate with 1 mL of Dulbecco’s modified eagle medium (DMEM, Gibco, Waltham, MA, USA) containing 2 mM of glutamine and 10% heat-inactivated fetal bovine serum (HI-FBS, Gibco). 

To evaluate the potential for inhibiting biofilm development on the skin, 50 µL of bacteria (1 × 10^6^ CFU/mL for *E. faecium*, or 1 × 10^5^ CFU/mL for *K. pneumoniae* or *E. cloacae*) plus broth containing farnesol at 1 or 6 mg/mL was spread onto the intact or burned skin. For the creation of established biofilms, followed by farnesol exposure, 50 µL of bacteria containing 1 × 10^8^ CFU/mL for *E. faecium* or 1 × 10^7^ CFU/mL for *K. pneumoniae* or *E. cloacae* was spread on the skin; then, the skin samples (with the dermis in media and epidermal surface exposed to air) were incubated at 37 °C and 5% CO_2_ for 24 h in a humid chamber. The treatment of 24 h old established biofilms involved the addition of 100 µL of farnesol at 15 mg/mL, or ethanol, in broth being drop deposited uniformly over the established biofilms on the skin samples, followed by incubation at 37 °C and 5% CO_2_ for 24 h. 

Three 5 mm punch biopsies were collected from the skin samples to aid in the determination of a viable bacterial count. Bacteria were collected from the biopsy surface by sterile swabbing (30 times). The swab head was then aseptically removed and added to a tube containing 1 mL of sterile PBS, and the tube was vortexed for 30 s. The number of viable bacteria over the skin surface area (CFU/cm^2^ of the skin surface) was determined by serial dilution, plating onto agar, incubation at 37 °C for 24 h, and visual counting of colonies. The lower limit of detection was 255 CFU/cm^2^ (=50 CFU/area in cm^2^ of a 0.5 cm punch biopsy), allowing a value of 255 CFU/cm^2^ to be assigned when no growth occurred. One additional 5 mm punch biopsy was collected for histological evaluation by fixing it in 10% formalin for 24 h, embedding in paraffin, sectioning at 5 µm, de-paraffinizing, staining with hematoxylin (1% for 3 min) and eosin (1% for 1 min), and image analysis using a Zeiss Axioscope microscope (Zeiss, Oberkochen, Baden-Württemberg, Germany).

### 4.9. Statistical Analysis

Data represent the mean ± standard deviation of the mean. The statistical evaluation was procured using GraphPad Prism 9 (version 9.2.0), with statistical tests being two-sided. Student’s t-test was used for a comparison between the two groups. The quantification of the means involving three or more groups employed analysis of variance (ANOVA) testing. When the ANOVA justified post hoc comparisons, the group comparisons were evaluated using Tukey’s multiple-comparisons test. The results were considered statistically significant at a value of *p* < 0.05.

## Figures and Tables

**Figure 1 antibiotics-13-00778-f001:**
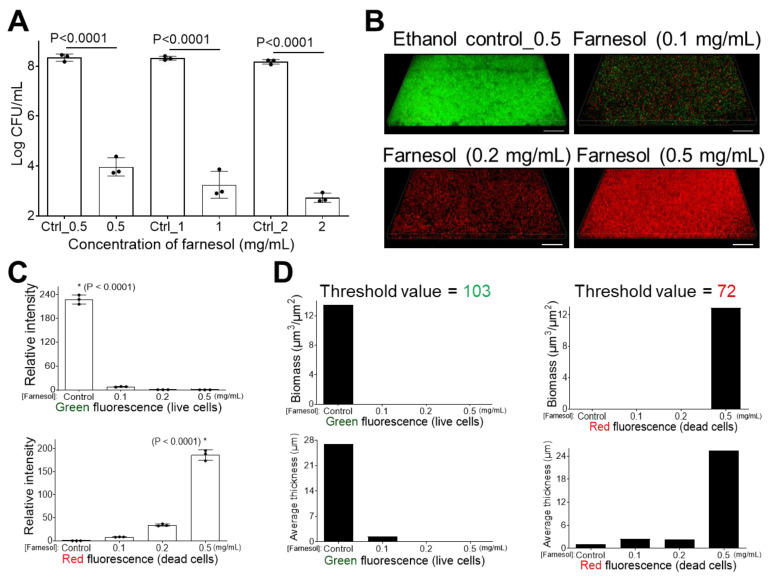
Inhibition of *E. faecium* biofilm formation. (**A**) *E. faecium* biofilm development is inhibited with exposure to farnesol for 24 h. Data are represented as the mean ± standard deviation (SD) (n = 3). (**B**) Flattened views of three-dimensional stacked Live/Dead images of *E. faecium* biofilms after 24 h exposure to farnesol. Green indicates live cells (SYTO^®^ 9 staining), while red indicates dead cells (propidium iodide staining). Scale bars are 20 μm. (**C**) Quantitative evaluation of fluorescence intensity by Photoshop^®^ from the images in (**B**), with data representing the mean ± SD (n = 3). (**D**) Quantitative evaluation of fluorescence intensity by Comstat2 from the images in (**B**) (biomass (µm^3^/µm^2^) and average thickness (µm)), using Otsu thresholding [31]. Ctrl indicates the ethanol control corresponding to the same volume of ethanol used to carry farnesol. Ctrl_0.5 = 1.7%, Ctrl_1 = 3.3%, and Ctrl_2 = 6.7% of ethanol. * *p* < 0.0001 against each of other groups.

**Figure 2 antibiotics-13-00778-f002:**
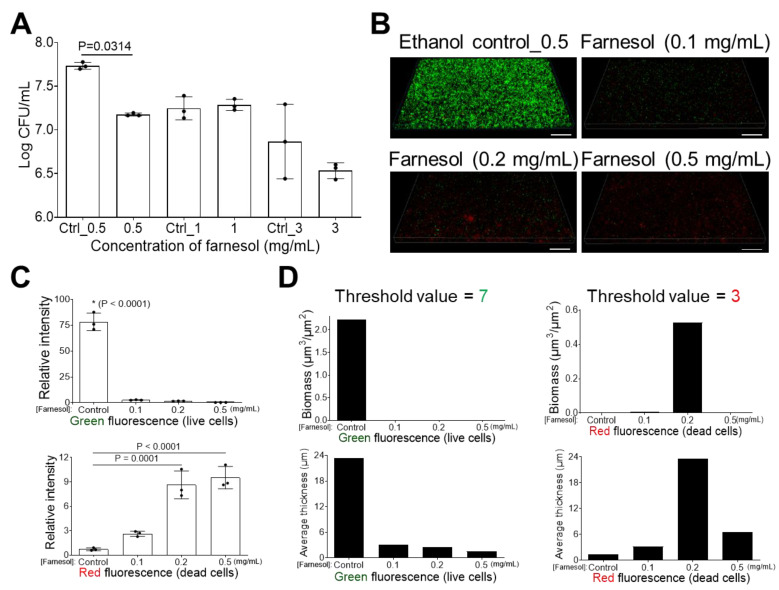
Inhibition of *K. pneumoniae* biofilm formation. (**A**) *K. pneumoniae* biofilm development is inhibited with exposure to farnesol for 24 h. Data are represented as the mean ± SD (n = 3). (**B**) Flattened views of three-dimensional stacked Live/Dead images of *K. pneumoniae* biofilms after 24 h exposure to farnesol. Green indicates live cells (SYTO^®^ 9 staining), while red indicates dead cells (propidium iodide staining). Scale bars are 20 μm. (**C**) Quantitative evaluation of fluorescence intensity by Photoshop^®^ from the images in (**B**), with data representing the mean ± SD (n = 3). (**D**) Quantitative evaluation of fluorescence intensity by Comstat2 from the images in (**B**) (biomass (µm^3^/µm^2^) and average thickness (µm)), using Otsu thresholding [31]. Ctrl indicates the ethanol control corresponding to the same volume of ethanol used to carry farnesol. Ctrl_0.5 = 1.7%, Ctrl_1 = 3.3%, and Ctrl_3 = 10% of ethanol. * *p* < 0.0001 against each of other groups.

**Figure 3 antibiotics-13-00778-f003:**
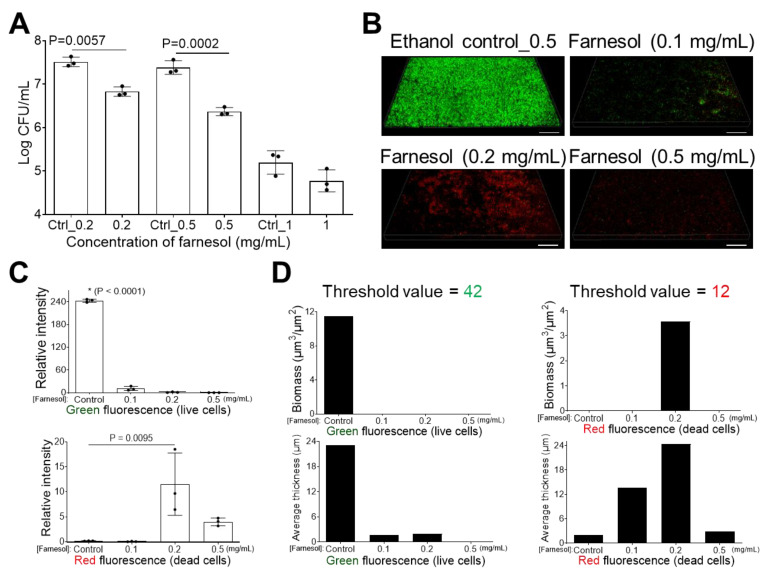
Inhibition of *E. cloacae* biofilm formation. (**A**) *E. cloacae* biofilm development is inhibited with exposure to farnesol for 24 h. Data are represented as the mean ± SD (n = 3). (**B**) Flattened views of three-dimensional stacked Live/Dead images of *E. cloacae* biofilms after 24 h exposure to farnesol. Green indicates live cells (SYTO^®^ 9 staining), while red indicates dead cells (propidium iodide staining). Scale bars are 20 μm. (**C**) Quantitative evaluation of fluorescence intensity by Photoshop^®^ from the images in (**B**), with data representing the mean ± SD (n = 3). (**D**) Quantitative evaluation of fluorescence intensity by Comstat2 from the images in (**B**) (biomass (µm^3^/µm^2^) and average thickness (µm)), using Otsu thresholding [31]. Ctrl indicates the ethanol control corresponding to the same volume of ethanol used to carry farnesol. Ctrl_0.2 = 0.67%, Ctrl_0.5 = 1.7%, and Ctrl_1 = 3.3% of ethanol. * *p* < 0.0001 against each of other groups.

**Figure 4 antibiotics-13-00778-f004:**
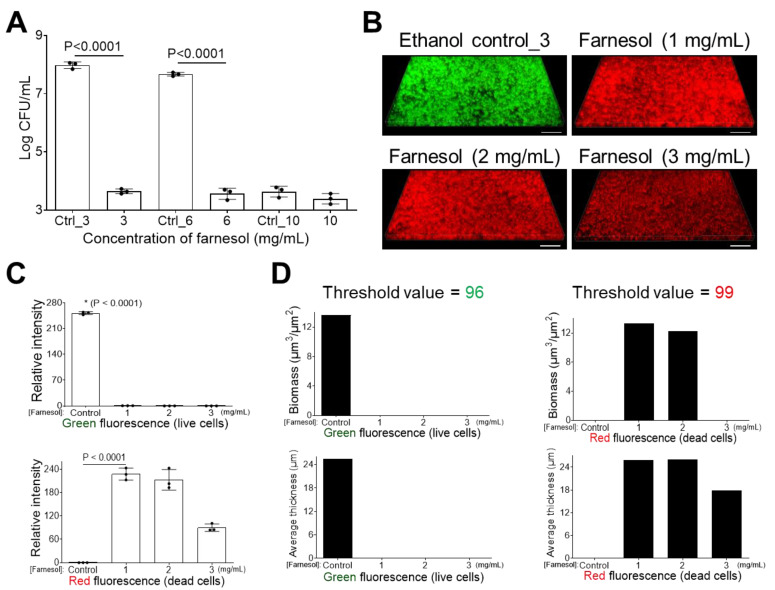
Farnesol disrupts established *E. faecium* biofilms. (**A**) The 24 h old established biofilms exposed to farnesol for 24 h have reduced cell viability, indicative of biofilm disruption and cell killing. Data are represented as the mean ± SD (n = 3). (**B**) Flattened views of three-dimensional stacked Live/Dead images of 24 h old established *E. faecium* biofilms after 24 h exposure to farnesol. Green indicates live cells (SYTO^®^ 9 staining), while red indicates dead cells (propidium iodide staining). Scale bars are 20 μm. (**C**) Quantitative evaluation of fluorescence intensity by Photoshop^®^ from the images in (**B**), with data representing the mean ± SD (n = 3). (**D**) Quantitative evaluation of fluorescence intensity by Comstat2 from the images in (**B**) (biomass (µm^3^/µm^2^) and average thickness (µm)), using Otsu thresholding [31]. Ctrl indicates the ethanol control corresponding to the same volume of ethanol used to carry farnesol. Ctrl_3 = 10%, Ctrl_6 = 20%, and Ctrl_10 = 33.3% of ethanol. * *p* < 0.0001 against each of other groups.

**Figure 5 antibiotics-13-00778-f005:**
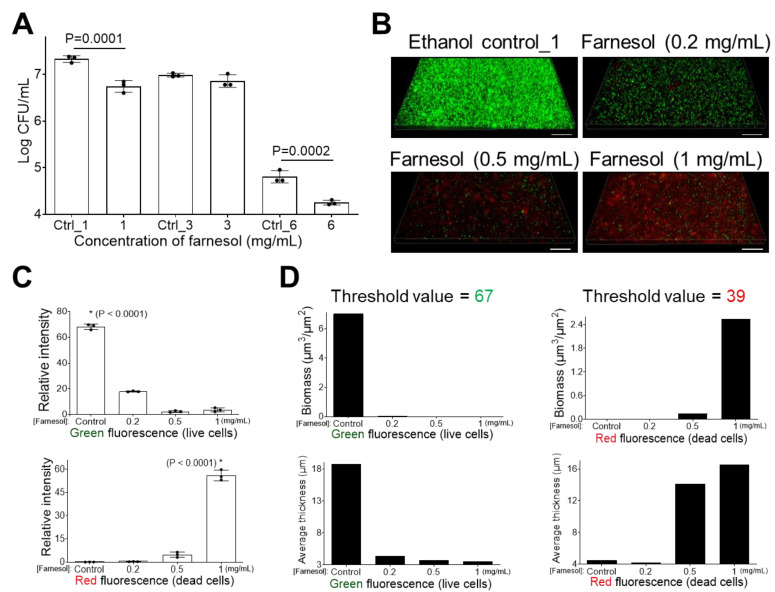
Farnesol disrupts established *K. pneumoniae* biofilms. (**A**) The 24 h old established biofilms exposed to farnesol for 24 h have reduced cell viability, indicative of biofilm disruption and cell killing. Data are represented as the mean ± SD (n = 3). (**B**) Flattened views of three-dimensional stacked Live/Dead images of 24 h old established *K. pneumoniae* biofilms after 24 h exposure to farnesol. Green indicates live cells (SYTO^®^ 9 staining), while red indicates dead cells (propidium iodide staining). Scale bars are 20 μm. (**C**) Quantitative evaluation of fluorescence intensity by Photoshop^®^ from the images in (**B**), with data representing the mean ± SD (n = 3). (**D**) Quantitative evaluation of fluorescence intensity by Comstat2 from the images in (**B**) (biomass (µm^3^/µm^2^) and average thickness (µm), using Otsu thresholding [31]. Ctrl indicates the ethanol control corresponding to the same volume of ethanol used to carry farnesol. Ctrl_1 = 3.3%, Ctrl_3 = 10%, and Ctrl_6 = 20% of ethanol. * *p* < 0.0001 against each of other groups.

**Figure 6 antibiotics-13-00778-f006:**
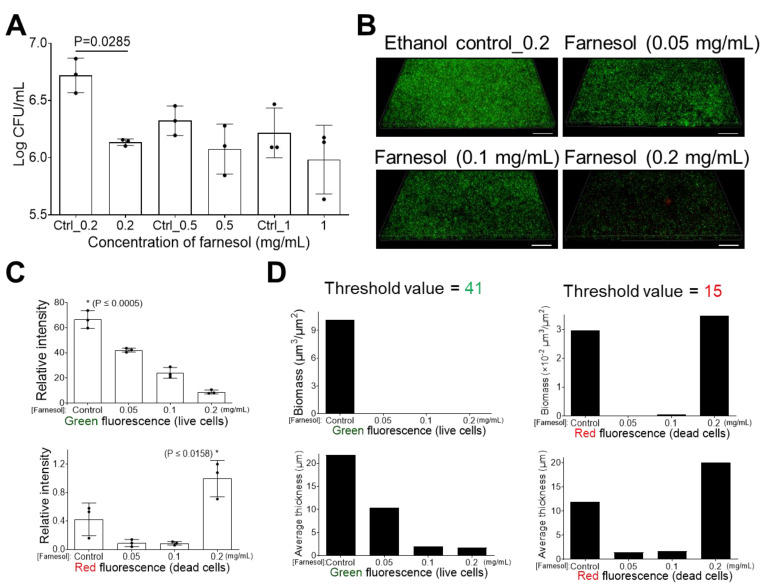
Farnesol disrupts established *E. cloacae* biofilms. (**A**) The 24 h old established biofilms exposed to farnesol for 24 h have reduced cell viability, indicative of biofilm disruption and cell killing. Data are represented as the mean ± SD (n = 3). (**B**) Flattened views of three-dimensional stacked Live/Dead images of 24 h old established *E. cloacae* biofilms after 24 h exposure to farnesol. Green indicates live cells (SYTO^®^ 9 staining), while red indicates dead cells (propidium iodide staining). Scale bars are 20 μm. (**C**) Quantitative evaluation of fluorescence intensity by Photoshop^®^ from the images in (**B**), with data representing the mean ± SD (n = 3). (**D**) Quantitative evaluation of fluorescence intensity by Comstat2 from the images in (**B**) (biomass (µm^3^/µm^2^) and average thickness (µm)), using Otsu thresholding [31]. Ctrl indicates the ethanol control corresponding to the same volume of ethanol used to carry farnesol. Ctrl_0.2 = 0.67%, Ctrl_0.5 = 1.7%, and Ctrl_1 = 3.3% of ethanol. * *p* ≤ 0.0005 against each of other groups.

**Figure 7 antibiotics-13-00778-f007:**
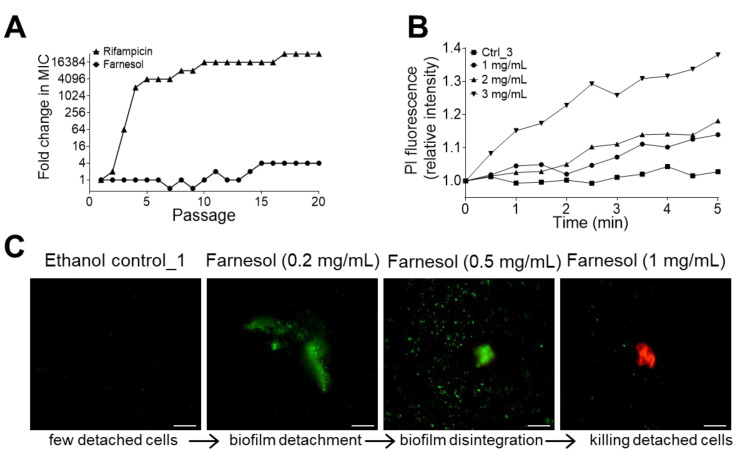
Farnesol kills *E. faecium* cells and facilitates biofilm detachment without inducing resistance. (**A**) *E. faecium* development of resistance to rifampicin or farnesol with serial passaging at sub-MIC doses. (**B**) Propidium iodide (PI) influx into *E. faecium* indicates cell killing. Data include the mean of three replicates. (**C**) The ethanol vehicle does not detach or kill *E. faecium* cells following 24 h exposure of established biofilms to the ethanol control, as indicated by no green or red fluorescence. Supernatants recovered from the biofilms indicate that live (green) biomass is detached from the surface at a low dose of farnesol. Increasing farnesol doses disrupt the live biomass and further kill the *E. faecium* cells (red floating material in the supernatant). Scale bars, 20 μm. Ctrl_1 = 3.3%, and Ctrl_3 = 10% of ethanol.

**Figure 8 antibiotics-13-00778-f008:**
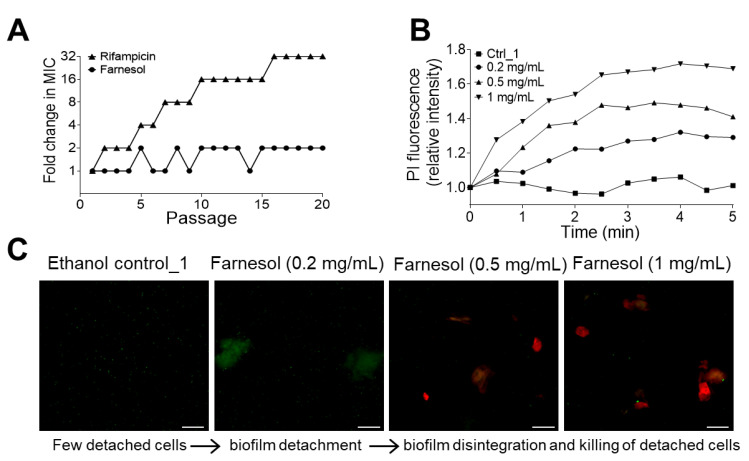
Farnesol kills *K. pneumoniae* cells and facilitates biofilm detachment without inducing resistance. (**A**) *K. pneumoniae* development of resistance to rifampicin or farnesol with serial passaging at sub-MIC doses. (**B**) Propidium iodide (PI) influx into *K. pneumoniae* indicates cell killing. Data include the mean of three replicates. (**C**) The ethanol vehicle does not detach or kill *K. pneumoniae* cells following 24 h exposure of established biofilms to the ethanol control, as indicated by no green or red fluorescence. Supernatants recovered from the biofilms indicate that live (green) biomass is detached from the surface at a low dose of farnesol. Increasing farnesol doses disrupt the live biomass and further kill the *K. pneumoniae* cells (red floating material in the supernatant). Scale bars, 20 μm. Ctrl_1 = 3.3% of ethanol.

**Figure 9 antibiotics-13-00778-f009:**
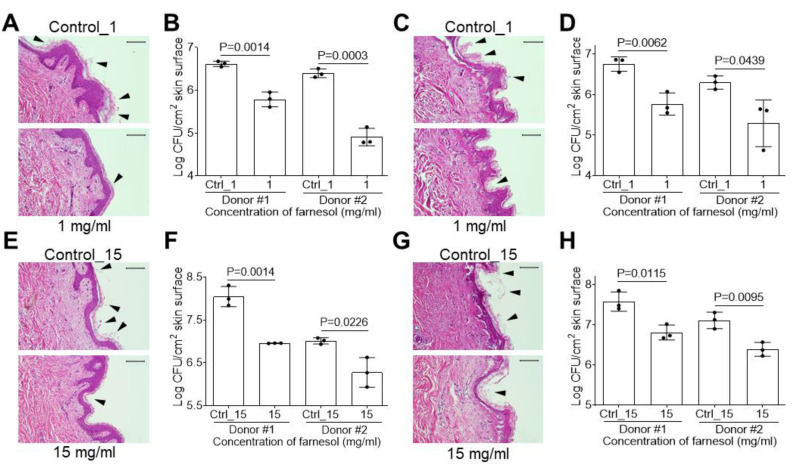
Farnesol prevents biofilm development and disrupts established *E. faecium* biofilm*s* on ex vivo intact or burned human skin. (**A**,**B**) *E. faecium* biofilm development on ex vivo human skin is inhibited by exposure to farnesol at 1 mg/mL for 24 h, as visualized by H&E-stained images (**A**), and quantification of viable bacteria in log_10_ CFU per square centimeter (cm^2^) of skin (**B**). (**C**,**D**) The same concentration of farnesol (1 mg/mL) is also effective for inhibiting *E. faecium* biofilm development on burned ex vivo human skin treated for 24 h, as visualized by H&E-stained images (**C**), and quantification of viable bacteria in log_10_ CFU per cm^2^ of skin (**D**). (**E**,**F**) Established *E. faecium* biofilms developed for 24 h on ex vivo human skin, and then exposed to farnesol for 24 h (15 mg/mL), have reductions in biofilm development, as visualized by H&E-stained images (**E**), and quantification of viable bacteria in log_10_ CFU per cm^2^ of skin (**F**). (**G**,**H**) The same concentration of farnesol (15 mg/mL) is also effective for inhibiting *E. faecium* biofilm development on burned ex vivo human skin treated for 24 h, as visualized by H&E-stained images (**G**), and quantification of viable bacteria in log_10_ CFU per cm^2^ of skin (**H**). The presence of biofilm is indicated by arrowheads in (**A**,**C**,**E**,**G**). Quantitative data in (**B**,**D**,**F**,**H**) represent the mean ± SD (n = 3 separate pieces of skin) from two independent donors. Ctrl_1 = 3.3 and Ctrl_15 = 50% of ethanol.

**Figure 10 antibiotics-13-00778-f010:**
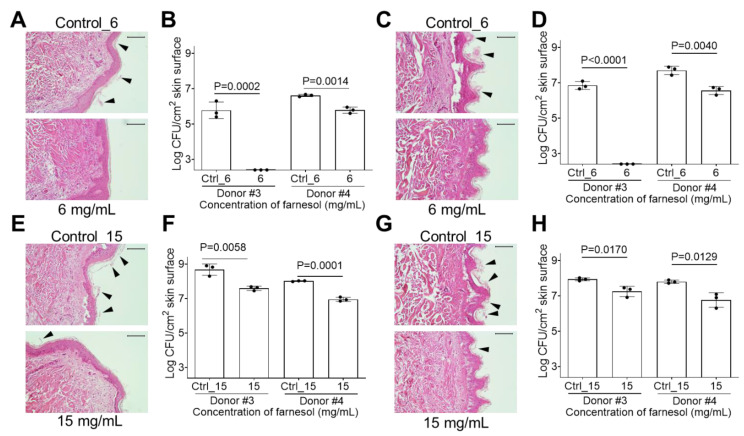
Farnesol prevents biofilm development and disrupts established *K. pneumoniae* biofilm*s* on ex vivo intact or burned human skin. (**A**,**B**) *K. pneumoniae* biofilm development on ex vivo human skin is inhibited by exposure to farnesol at 1 mg/mL for 24 h, as visualized by H&E-stained images (**A**), and quantification of viable bacteria in log_10_ CFU per cm^2^ of skin (**B**). (**C**,**D**) The same concentration of farnesol (1 mg/mL) is also effective for inhibiting *K. pneumoniae* biofilm development on burned ex vivo human skin treated for 24 h, as visualized by H&E-stained images (**C**), and quantification of viable bacteria in log_10_ CFU per cm^2^ of skin (**D**). (**E**,**F**) Established *K. pneumoniae* biofilms developed for 24 h on ex vivo human skin, and then exposed to farnesol for 24 h (15 mg/mL), have reductions in biofilm development, as visualized by H&E-stained images (**E**), and quantification of viable bacteria in log_10_ CFU per cm^2^ of skin (**F**). (**G**,**H**) The same concentration of farnesol (15 mg/mL) is also effective for inhibiting *K. pneumoniae* biofilm development on burned ex vivo human skin treated for 24 h, as visualized by H&E-stained images (**G**), and quantification of viable bacteria in log_10_ CFU per cm^2^ of skin (**H**). The presence of biofilm is indicated by arrowheads in (**A**,**C**,**E**,**G**). Quantitative data in (**B**,**D**,**F**,**H**) represent the mean ± SD (n = 3 separate pieces of skin) from two independent donors. Ctrl_6 = 20% of ethanol; Ctrl_15 = 50% of ethanol.

**Figure 11 antibiotics-13-00778-f011:**
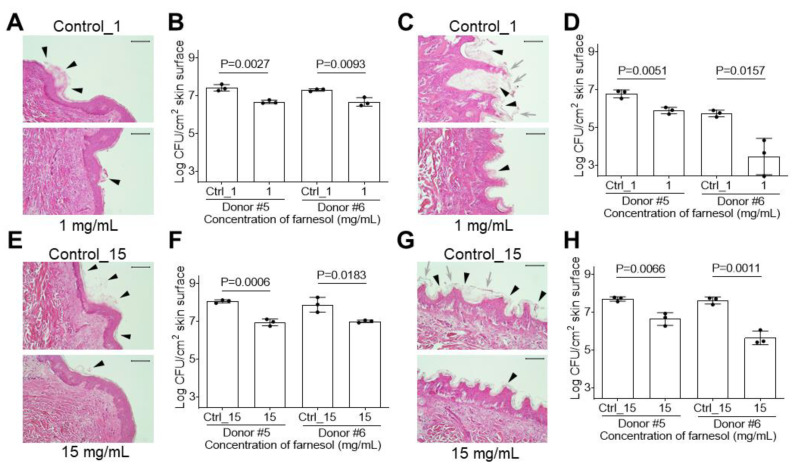
Farnesol prevents biofilm development and disrupts established *E. cloacae* biofilm*s* on ex vivo intact or burned human skin. (**A**,**B**) *E. cloacae* biofilm development on ex vivo human skin is inhibited by exposure to farnesol at 1 mg/mL for 24 h, as visualized by H&E-stained images (**A**), and quantification of viable bacteria in log_10_ CFU per cm^2^ of skin (**B**). (**C**,**D**) The same concentration of farnesol (1 mg/mL) is also effective for inhibiting *E. cloacae* biofilm development on burned ex vivo human skin treated for 24 h, as visualized by H&E-stained images (**C**), and quantification of viable bacteria in log_10_ CFU per cm^2^ of skin (**D**). (**E**,**F**) Established *E. cloacae* biofilms developed for 24 h on ex vivo human skin, and then exposed to farnesol for 24 h (15 mg/mL), have reductions in biofilm development, as visualized by H&E-stained images (**E**), and quantification of viable bacteria in log_10_ CFU per cm^2^ of skin (**F**). (**G**,**H**) The same concentration of farnesol (15 mg/mL) is also effective for *E. cloacae* biofilm development on burned ex vivo human skin treated for 24 h, as visualized by H&E-stained images (**G**), and quantification of viable bacteria in log_10_ CFU per cm^2^ of skin (**H**). The presence of biofilm is indicated by black arrowheads in (**A**,**C**,**E**,**G**). The gray arrows in (**C,G**) indicate broken stratum corneum of burned epidermis, which was penetrated by *E. cloacae* to develop biofilm. Quantitative data in (**B**,**D**,**F**,**H**) represent the mean ± SD (n = 3 separate pieces of skin) from two independent donors. Ctrl_1 = 3.3% of ethanol; Ctrl_15 = 50% of ethanol.

## Data Availability

The datasets generated during and/or analyzed during the current study are available from the corresponding author on request.

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
