# Peer review of "Farnesol Emulsion as an Effective Broad-Spectrum Agent against ESKAPE Biofilms"

_antibiotics, 2024, doi:10.3390/antibiotics13080778_

Round 1

Reviewer 1 Report

Comments and Suggestions for Authors

The manuscript presented for review is well-written and structured article about the ESCAPE pathogens and the ability of farnesol, as an Agent Against ESKAPE biofilms. Due to the increasing resistance of ESCAPE pathogens, including those capable of causing nosocomial infections, the topic raised is an important clinical problem.  Although the subject of the work is interesting, the manuscript has a few errors which are listed below.

Minor specific suggestion/comments: 

- Lines 10,11,16,18 and others - italics

- Please change the units throughout the article - mg/mL, not mg/ml.

- Lines 161, 166 - Klebsiella pneumoniae

- Line 359 - Enterobacter spp.

- Figures 9,10,11 - Wouldn't it be better to standardize the scale on the OY axis for parts B, D, F, H - the same for all?

- Line 464 - "and" not italics.

Author Response

Reviewer 1: The manuscript presented for review is well-written and structured article about the ESCAPE pathogens and the ability of farnesol, as an Agent Against ESKAPE biofilms. Due to the increasing resistance of ESCAPE pathogens, including those capable of causing nosocomial infections, the topic raised is an important clinical problem.  Although the subject of the work is interesting, the manuscript has a few errors which are listed below.

Minor specific suggestion/comments: 

1 Lines 10,11,16,18 and others – italics

Response: We apologize for this oversight.  All bacteria names throughout the manuscript are now presented in italics

  1. Please change the units throughout the article - mg/mL, not mg/ml.

Response: Thank you for identifying this problem in notation.  Changes have been made throughout the manuscript.

  1. Lines 161, 166 - Klebsiella pneumonia

Response: Thank you for pointing out this error in spell-checking.  Pneumonia has been changed to pneumoniae throughout the manuscript, in addition to the specific lines noted.

  1. Line 359 - Enterobacterspp.

Response: Thank you for identifying this error.  We have corrected the bacteria name and used species instead of spp.

  1. Figures 9,10,11 - Wouldn't it be better to standardize the scale on the OY axis for parts B, D, F, H - the same for all?

Response: Thank you for this suggestion.  Figures 9,10, and 11 have all been standardized so that the y-axis ranges from 2.4 to 9.4.

  1. Line 464 - "and" not italics.

Response: Thank you for identifying this problem. The italics have been removed.

Reviewer 2 Report

Comments and Suggestions for Authors

ESKAPE pathogens are bacteria that cause hard-to-treat hospital infections and resist common antibiotics. Farnesol, an ingredient safely used in cosmetics, can effectively break down and kill the biofilms these bacteria form. It works against all six ESKAPE pathogens and even prevents them from developing resistance. Farnesol was also safe and effective in treating these infections on human skin models, making it a promising new treatment option.

There are numerous concerns should be considered before publication:

1. Please avoid the same words for keywords that you already used for title.

2. Abstract is not clear and doesn’t provide sufficient information about the work done. Please rewrite.

3. Line 57: please don’t use active English structure. Please check throughout the article considering this point.

4. Fig. 7: Where are the standard deviations of the results? Please put them on the figure.

5. Fig. 8: Where are the standard deviations of the results? Please put them on the figure.

6. Conclusion is recommended to write by highlighting the works core results.

7. Similarity index is very high.

8. From the linguistic point of view, the article should be rewritten.

Comments on the Quality of English Language

Moderate editing of English language required.

Author Response

Reviewer 2: ESKAPE pathogens are bacteria that cause hard-to-treat hospital infections and resist common antibiotics. Farnesol, an ingredient safely used in cosmetics, can effectively break down and kill the biofilms these bacteria form. It works against all six ESKAPE pathogens and even prevents them from developing resistance. Farnesol was also safe and effective in treating these infections on human skin models, making it a promising new treatment option. There are numerous concerns should be considered before publication:

  1. Please avoid the same words for keywords that you already used for title.

Response: The keywords have been changed so they differ from words in the title.

  1. Abstract is not clear and doesn’t provide sufficient information about the work done. Please rewrite.

Response: We apologize that the abstract was not clear.  The abstract has been re-written for clarity and also to provide more details, as requested by reviewer 3.

  1. Line 57: please don’t use active English structure. Please check throughout the article considering this point.

Response: Thank you for pointing this out.  We have thoroughly checked the manuscript and revised any usage of active tense.

  1. Fig. 7: Where are the standard deviations of the results? Please put them on the figure.

Response: Due to the nature of the experiment, the concentration of farnesol or antibiotic that was effective for inhibiting biofilm growth was determined visually, not quantitatively.  We have revised the manuscript to explain this in the results and refer the reader to the methods section 4.6 where the experimental design was described. Therefore, there are not error bars that can be added for each point of the graph.

  1. Fig. 8: Where are the standard deviations of the results? Please put them on the figure.

Response: Due to the nature of the experiment, the concentration of farnesol or antibiotic that was effective for inhibiting biofilm growth was determined visually, not quantitatively.  We have revised the manuscript to explain this in the results and refer the reader to the methods section 4.6 where the experimental design was described. Therefore, there are not error bars that can be added for each point of the graph.

  1. Conclusion is recommended to write by highlighting the works core results.

Response: The conclusion has been re-written for clarity and to highlight the key results.

  1. Similarity index is very high.

Response: The manuscript has been checked and passages that arte similar with our previous work have been revised to minimize similarity index.

  1. From the linguistic point of view, the article should be rewritten.

Response: We apologize that the manuscript language was not clear.  We have thoroughly reviewed the manuscript to improve clarity and English usage.

Reviewer 3 Report

Comments and Suggestions for Authors

The manuscript presents important results of farnesol in the control of bacterial infections, especially biofilms. however, before publication, it requires few corrections.

ABSTRACT 

1- Please, put the names of microorganisms in italics;

2- Briefly add the methodology used and the main results. This information is important for readers to have initial knowledge of the manuscript.

INTRODUCTION

1- Add a paragraph associating the recent SARS-Cov-2 pandemic and the increase in bacterial infections around the world, especially ESKAPE.

MATERIAL AND METHODS 

1- Item 4.3. Inhibiting biofilm formation: 

What was the concentration of farnesol?

In addition to the ethanol control, was there a negative treatment control? Which?

2- Item 4.8. Developing and treating biofilms on ex vivo human skin:

How long was the formalin fixation time? What dyes were used for histology?

RESULTS 

1- line 127

"but most of the dead cells failed to attach to the surfaces at 0.5 mg/ml of farnesol (Figure 3B-D"

What could have happened? Was the concentration too high? If this is confirmed, I suggest leaving only concentrations of 0.1 and 0.2 mg/ mL. This goes for the other strains.

2- Is it possible to improve histology images? It is not possible to observe biofilms.

Author Response

Reviewer 3: The manuscript presents important results of farnesol in the control of bacterial infections, especially biofilms. however, before publication, it requires few corrections.

ABSTRACT 

1- Please, put the names of microorganisms in italics.

Response: We apologize for the mistakes, and the names of all microorganisms are identified in italics throughout the revised manuscript.

2- Briefly add the methodology used and the main results. This information is important for readers to have initial knowledge of the manuscript.

Response: Thank you for the helpful suggestion. The abstract has been re-written for clarity as per the suggestion of reviewer 2, and the methods and key findings included. 

INTRODUCTION

1- Add a paragraph associating the recent SARS-Cov-2 pandemic and the increase in bacterial infections around the world, especially ESKAPE.

Response: Thank you for pointing out the need to include the relevant information on ESKAPE pathogen infections associated with SARS-Covid19.  We have added a new paragraph in the introduction, with the inclusion of beneficial references.

MATERIAL AND METHODS 

1- Item 4.3. Inhibiting biofilm formation: 

What was the concentration of farnesol?

Response: We apologize that the range of concentrations of farnesol was not added in the methods.  Sections 4.3, 4.4 and 4.8, for developing and established biofilm, have been revised to include the range of concentrations used, as these differ.  We hope this inclusion clarifies the treatment parameters and aids the reader’s understanding of the results.

2- In addition to the ethanol control, was there a negative treatment control? Which?

Response: We thank the reviewer for the suggestion.  Since ethanol was the carrier solvent for creation of the farnesol emulsion and farnesol is not soluble in aqueous media we did not include an additional negative control of water alone.  We have revised section 4.3 to better explain the rationale for this decision.

3- Item 4.8. Developing and treating biofilms on ex vivo human skin:

How long was the formalin fixation time? What dyes were used for histology?

Response: We apologize for not providing sufficient details. This section has been revised to include that the time for fixation was 24 hr, the sections de-paraffinized, and the hematoxylin and eosin stains were used at a 1% solution and applied for 3 and 1 min with water and ethanol washing steps

RESULTS 

1- line 127

"but most of the dead cells failed to attach to the surfaces at 0.5 mg/ml of farnesol (Figure 3B-D"

What could have happened? Was the concentration too high? If this is confirmed, I suggest leaving only concentrations of 0.1 and 0.2 mg/ mL. This goes for the other strains.

Response: We thank the reviewer for consideration of this point. Farnesol at lower concentrations appears to halt biofilm adherence, as shown in the data in figures 7 and 8.  We have revised the text explain the results in Figure 2 and 3 where the red ‘dead’ signal is not observed and why.  We also refer the reader to the results of Figured 7 and 8 where we sought to demonstrate that bacterial cells are removed from the biofilm with the application of low doses of farnesol, yet they remain viable unless higher concentrations of farnesol are used to kill the bacteria.

2- Is it possible to improve histology images? It is not possible to observe biofilms.

Response: We thank the reviewer for this suggestion.  However, we do not have alternative histology images and feel that the images provided demonstrate the presence, or lack of biofilm with farnesol treatment. An important feature of the images is that they also show that farnesol application does not cause damage to the skin, which is critical for safe application. The images are similar to those published by other teams, and we have included arrows to indicate where biofilm can be seen.  We have also modified the text to help the reader understand that the arrows indicate the presence of biofilm. 

Round 2

Reviewer 2 Report

Comments and Suggestions for Authors

From my side, the article can be published as is.

Comments on the Quality of English Language

Minor editing of English language required.